# Comparison of Effects of Storage at Different Temperatures in a Refrigerator, Upright Freezer on Top of Refrigerator, and Deep Freezer on the Immunoglobulin A Concentration and Lysozyme Activity of Human Milk

**DOI:** 10.3390/ijerph192013203

**Published:** 2022-10-13

**Authors:** Xuejing Li, Penprapa Siviroj, Jetsada Ruangsuriya, Chotiros Phanpong, Wachiranun Sirikul, Krongporn Ongprasert

**Affiliations:** 1Department of Community Medicine, Faculty of Medicine, Chiang Mai University, Chiang Mai 50200, Thailand; 2Department of Biochemistry, Faculty of Medicine, Chiang Mai University, Chiang Mai 50200, Thailand; 3Regional Health Promotion Center 1, Chiang Mai 50100, Thailand

**Keywords:** bioactive milk molecules, breast milk, innate immunity, human milk, storage

## Abstract

This study aimed to investigate the effects of storing expressed human milk (HM) at different domestic storage temperatures on the secretory immunoglobulin A (SIgA) concentration and lysozyme activity. Forty mothers of full-term infants aged one to six months provided milk samples. The fresh samples were examined within 24 h of expression, and the other samples were stored in a refrigerator for four days or in two types of freezers for six months. The SIgA concentrations and lysozyme activity in the milk samples were studied using enzyme-linked immunosorbent assay (ELISA) kits and fluorometric lysozyme activity assay kits, respectively. The pairwise comparisons of the SIgA concentration and lysozyme activity were carried out using one-way analysis of variance with Dunnett T3 or Kruskal–Wallis tests with Bonferroni correction, depending on the data distribution. The mean temperatures of the refrigerator, upright freezer on top of the refrigerator, and deep freezer (chest freezer) were 2.0, −16.7, and −22.3 °C, respectively. Our study results highlight that the SIgA concentration and lysozyme activity of HM stored in the refrigerator for four days and in freezers for six months were significantly lower than those of fresh HM (*p* < 0.001). During the first six months of storage in both types of freezers, the SIgA levels were stable, whereas the lysozyme activity significantly decreased (*p* < 0.001). HM stored in the deep freezer had a higher SIgA concentration and lysozyme activity than HM stored in the upright freezer on top of the refrigerator. Our data support the superiority of fresh human milk over stored HM. If HM is to be stored, then storage in a deep freezer is potentially a more effective method for the preservation of SIgA concentrations and lysozyme activity than storage by refrigeration for four days or in an upright freezer on top of a refrigerator for six months.

## 1. Introduction

Expressed human milk (HM) is critical for infants if they are to benefit from their mother’s milk when they are unable to breastfeed directly, and therefore methods have been developed so they can be fed with expressed HM. The need for this may arise because of the condition of the mother (e.g., varicella, tuberculosis, or COVID-19 infection) [1,2] or the infant, including defective, abnormal, or underdeveloped control of oropharyngeal structures, resulting in ineffective sucking (e.g., in very preterm infants, cases of hypotonia, or infants with a cleft lip, cleft palate, or both) [3,4,5,6]. Expressed HM also has advantages in a range of modern sociocultural contexts, which has increased the popularity of prolonged breastfeeding while introducing complementary foods for at least one to two years, as recommended by major international health authorities [7,8], in addition to increasing awareness of supporting mothers to persist in breastfeeding after returning to school or work [9,10,11]. In addition, because of the extension of LGBTQIA+ rights and the legalization of same-sex marriage [12], HM expression may be considered if latching on is not desired or if the infant is not able to latch on, e.g., in the case of a mother who is transgender [13]. According to the 2019 Thailand Multiple Indicator Cluster Survey, the proportion of exclusive and predominant breastfeeding for six months was 14% and 31%, respectively. The proportion of children aged between 13 and 15 months who were breastfed was 25%, and for those between 20 and 23 months it was 15%. This report included all methods of breastmilk feeding, namely, direct nursing, expressing, and bottle or cup feeding [14]. According to studies conducted in Perth, Australia, the number of mothers who expressed breast milk increased from 38% to 69% between 1992 and 2002 [9].

Exploration of the effect of cooling conditions on the protein component of expressed HM stored for home usage is essential. Several milk proteins function to support infants’ innate immunity, which is a unique property that is not supported by infant formula [15] but may be lost after the storage of HM [16,17]. Our study here focuses on secretory immunoglobulin A (SIgA) concentrations and lysozyme activity in HM. SIgA is considered most important in terms of its concentration and biological properties in all stages of lactation [18,19]. SIgA in HM provides infants with passive immunity by neutralizing toxins and inhibiting pathogen adhesion to epithelial cells, and it is essential for protecting mucosal surfaces (gastrointestinal, respiratory) [18,20,21]. Lysozymes are one of the main enzymes found in HM, which contains significantly higher levels than bovine milk [22]. Lysozymes possess bactericidal properties through degradation of cell wall peptidoglycan in bacteria [23]. Additionally, lysozymes exhibit antifungal and antiviral properties [18,20]. Previous research on the effect of storage conditions on the SIgA concentrations and lysozyme activity is limited, and inconsistent results were reported, potentially because of differences in sample size, storage duration, and milk collection and preparation processes. For example, some studies collected and compared SIgA baseline data from individual HM samples with those from frozen milk samples obtained by pooling HM from a milk bank [24,25], whereas Ahrabi et al. [26] compared the IgA level in frozen samples with baseline samples from the same group of participants. In addition, in previous studies, milk samples have been collected at different times and stored in a laboratory freezer (−80 °C) until analysis, causing each sample to be exposed to extremely low temperatures for varying amounts of time before they were analyzed [25,26,27].

Recommendations for optimal HM storage conditions are based more on the colony-forming number than on the effect of milk storage on bioactive immunological components [28]. Storage recommendations vary from country to country. The majority of protocols accept storage in a refrigerator (39 °F/4 °C) for up to four days, in upright freezers on top of the refrigerator (0 °F/−18 °C or colder) for up to six months, and in deep freezers (−4 °F/−20 °C or colder) for up to 12 months (Table 1). In the earlier literature, a wide storage temperature range has been explored, from storing HM in a household freezer (−18 °C) to a laboratory freezer (−80 °C) [24,25,26,27,29,30]. Data from the previous literature are scarce, however, regarding storage temperatures corresponding to household conditions.

Our study was designed to examine the effect of cooling storage on the SIgA concentration and lysozyme activity in HM under domestically available storage conditions, including in refrigerators, upright freezers on top of refrigerators, and deep freezers (chest freezers). Our results may help identify the optimal storage conditions for the preservation of HM immunogenicity and in planning health actions aimed at maximizing human milk immunogenicity.

## 2. Materials and Methods

### 2.1. Setting

A recruitment poster was distributed in the well-child clinics and nursing rooms of three hospitals in the municipality of Chiang Mai, Thailand. All mothers provided breast milk samples at the Mother and Child Hospital in Chiang Mai, Thailand. Details on the sample size, participant recruitment, milk collection, and acquisition of milk samples have been published elsewhere [35].

### 2.2. Participants

Interested participants contacted the study personnel via telephone as indicated on the recruitment posters. The participants were questioned based on the eligibility criteria. This study recruited healthy breastfeeding mothers aged between 18 and 40 who had delivered a healthy full-term baby between 1 and 6 months ago and excluded those who were unable to independently visit our lactation room. The participants completed an informed consent form, and their travel expenses were reimbursed before they provided information and breast milk samples.

### 2.3. Milk Collection and Acquisition of Milk Samples

To minimize diurnal variation, all mothers were scheduled to provide breast milk samples on the same day between 08:00 and 11:00. The milk pump (Lactina Electric Selection Pump^®^, Medela Inc., Barr, Switzerland) was placed on the left and right breasts for 15 min or until no more milk was expressed for a minimum of 5 min to ensure that samples contained both fore and hind milk. After the expression process, milk samples were placed in milk bottles and immediately mixed by rotating while the milk was unsettled. The samples (120 mL) were then poured into three 50 mL polypropylene centrifuge tubes (NuncTM^®^, Roskilde, Denmark), of which 80 mL was analyzed for this study and the remaining 40 mL had been used for analysis in our earlier study [35]. HM samples were stored in an insulated container and aliquoted within four hours after collection. The samples were divided into 10 mL aliquots and poured into 15 mL polypropylene centrifuge tubes (NuncTM^®^, Roskilde, Denmark) that met the storage conditions specified in the protocol. Two 10 mL aliquots were refrigerated and analyzed within 24 h to identify the baseline SIgA levels and lysozyme activity (fresh), and these baseline results have been published elsewhere [35]. Two 10 mL aliquots were placed in a refrigerator for four days. Two 10 mL aliquots were frozen in an upright freezer on top of a refrigerator for six months. Two 10 mL aliquots were frozen in a deep freezer for six months (Figure 1).

### 2.4. Monitoring of Storage Temperatures

The storage temperatures were continuously monitored using a digital thermometer (Elitech, Modal RC-4, Guangzhou, China) and recorded using ElitechLogWIn V6.3.0. The mean (±2SD) temperatures were 2.0 ± 0.8 °C (range: 0.8 to 5.3 °C) for the refrigerator, −16.7 ± 0.9 °C (range: −12.2 to −18.4 °C) for the upright freezer on top of the refrigerator, and −22.3 ± 1.0 °C (range: −20.5 to −26.4 °C) for the deep freezer.

### 2.5. Thawing and Warming Processes

The frozen samples were thawed by transferring the samples from the freezers into a water bath (Memmert GmbH + Co.KG., Schwabach, West Germany) at 25 °C and incubating for 15 min. A digital thermometer (DeltaTrak^®^, Model 13309, Pleasanton, CA, USA) was used to monitor the temperature throughout the thawing and warming processes.

### 2.6. Analytical Methods

#### 2.6.1. SIgA Levels

ELISA kits (Aviva System Biology, cat no. OKEH00516, San Diego, CA, USA) were used to determine the SIgA levels in all milk samples. Following the manufacturer’s protocol, the milk samples were subjected to 200,000× dilution with deionized (DI) water and the assay diluent buffer. Next, 100 μL of the diluted samples, along with the SIgA standard, was added to each well of the ELISA plate. The plate was incubated at 37 °C for 120 min, and the solution in each well was replaced with the antibody detection solution. The plate was again incubated at 37 °C for 60 min, and, after a thorough wash, the solution in each well was replaced with the avidin-HRP conjugate mixture. The plate was again incubated at 37 °C for another 60 min, and the TMB substrate solution was added after a thorough wash. Next, the plate was incubated in the dark at 37 °C for 15 min, and stop solution was then added to stop the reaction. The absorbance at 450 nm was recorded using a Synergy H4 Hybrid Reader (Bio-Tek, Winooski, VT, USA), and the SIgA levels in the milk samples were extrapolated from the SIgA standard curve (0–4000 pg/mL).

#### 2.6.2. Lysozyme Activity

Commercial fluorometric lysozyme activity assay kits (MyBioSource Elabscience^®^, cat no. MBS846601, San Diego, CA, USA) were used to determine the activity of lysozyme in the milk samples, which were subjected to 20,000× dilution with DI water and the assay diluent before being loaded into each well of a 96-well plate. Subsequently, synthetic substrate was added, and the enzymatic reaction was allowed to proceed at 37 °C for 180 min. Finally, the stop solution was added to each, and the fluorescence intensities from each were recorded with a Synergy H4 Hybrid Reader (Bio-Tek, Winooski, VT, USA) using 360 nm excitation and 445 nm emission (Ex/Em = 360/445 nm) wavelengths. The lysozyme activity of each milk sample was calculated from the rate of 4-methylumbelliferone (4-MU) products using a standard curve (0 to 100 pmol/well) and expressed as nmol/min/mg of protein.

#### 2.6.3. Total Protein

Lowry’s method with Folin–Ciocalteu solution (VWR Chemicals, cat no. 31360.264, Radnor, PA, USA) was used to determine the total protein level in the milk samples. Each milk sample was subjected to 100× dilution with DI H_2_O and an alkaline solution, and Folin–Ciocalteu solution was added to the diluted sample, followed by thorough mixing. The mixture was incubated at 25 °C for 10 min, and the absorbance was read at 650 nm using a Synergy H4 Hybrid Reader (Bio-Tek, Winooski, VT, USA). As the standard, 0 to 100 mg/mL of bovine serum albumin (GE healthcare, cat no. K41-001, Chicago, IL, USA) was used, and the total protein content of each milk sample was calculated from the standard curve.

### 2.7. Ethical Considerations

Approval for this study was obtained from the Research Ethics Committee of the Faculty of Medicine, Chiang Mai University (no. 078/2021). All participants provided written informed consent. This study complied with the Declaration of Helsinki (1964) and all of its subsequent amendments.

### 2.8. Statistical Analysis

The STATA program (Stata Corp. 2019, Stata Statistical Software: Release 16, Stata Corp. LLC, College Station, TX, USA) was used to conduct all statistical analyses. The Shapiro–Wilk test was used to evaluate the normality of all variables. Outliers were detected by constructing boxplots of the SIgA concentration and lysozyme activity to remove extreme values from the data. For categorical data, the participant characteristics are reported as frequencies and with percentages. For parametric data, the mean with standard deviation (SD) or the 95% confidence interval (95% CI) were used, whereas for nonparametric data the median and 25th and 75th percentiles were employed. Comparison of the SIgA and lysozyme activity levels in fresh HM and in HM samples after storage in a refrigerator for four days, upright freezer on top of a refrigerator for six months, and deep freezer for six months was conducted via one-way analysis of variance (ANOVA) with Dunnett T3, and Kruskal–Wallis test with Bonferroni correction was applied for pairwise comparisons. Changes in the SIgA levels and lysozyme activities in HM stored in the deep freezer and in the upright freezer on top of the refrigerator during the six-month period were assessed using a mixed-effect regression model adjusted for time effects because there were repeated measurements. The statistical tests were all two-sided, and a *p*-value of 0.05 was considered to indicate statistically significant differences.

## 3. Results

### 3.1. Characteristics of the Participants

Forty mothers of full-term infants provided freshly expressed breast milk. The characteristics of the mothers and their children are shown in Table 2. The mean maternal age was 28.6 years and mean BMI was 23.8 kg/m^2^. The average postpartum age at milk collection was 3.31 months.

### 3.2. Comparison of the Effects of Different Storage Durations and Temperatures on the SIgA Levels

A comparison of the SIgA levels between different storage durations and temperatures is shown in Figure 2 and Table 3. The mean SIgA levels in fresh HM, samples stored in refrigerators (2 °C) for four days, in upright freezers on top of refrigerators (−16.7 °C), or in deep freezers (−22.3 °C) for six months were 27.329, 17.426, 24.077, and 27.178 mg/dL, respectively. Compared with fresh HM, the SIgA concentrations of the milk samples stored in refrigerators for four days showed the greatest reduction (36.23% decrease), which was statistically significant (*p* < 0.001). SIgA levels were lower in HM stored in freezers on top of refrigerators and in deep freezers for six months compared with fresh HM, but the difference was not statistically significant (*p* = 0.587 and 1.000, respectively). The best preservation of SIgA levels compared with fresh HM was observed after six months of deep freezing (0.60% decrease). A comparison of the SIgA levels between the three different storage methods revealed that the SIgA concentration was preserved at a higher level in the samples stored in an upright freezer on top of a refrigerator or in a deep freezer for six months than in a refrigerator for four days (*p* = 0.001 and < 0.001, respectively).

### 3.3. Comparison of the Effects of Storage Durations and Temperatures on Lysozyme Activity

A comparison of lysozyme activity levels for different storage durations and temperatures is shown in Figure 3 and Table 4. The mean lysozyme activities in fresh HM and in HM samples after storage in refrigerators (2 °C) for four days, upright freezers on top of refrigerators (−16.7 °C) for six months, and deep freezers (−22.3 °C) for six months were 0.923, 0.459, 0.391, and 0.564 nmol/mg protein, respectively. The lysozyme activity in HM was reduced after all three storage processes compared with the fresh HM samples (*p* < 0.001, Figure 3). Lysozyme activity in the milk samples stored in upright freezers on top of refrigerators for six months demonstrated the greatest reduction (48.65% decrease), which corresponded closely to storage in refrigerators for four days (44.64% decrease). Of the tested methods, storage in a deep freezer for six months resulted in the best preservation of, or lowest decrease in, lysozyme activity in the samples when compared with fresh HM (38.89% decrease). Frozen storage in upright freezers on top of refrigerators for six months showed lower preservation of lysozyme activities than storage in refrigerators for four days (*p* = 0.049). In a comparison of the lysozyme activities for storage in either type of freezer for six months, lysozyme activity was more effectively maintained in the samples stored in deep freezers than in upright freezers on top of refrigerators (*p* < 0.001).

### 3.4. Effects of Different Storage Durations and Temperatures on the SIgA Concentrations and Lysozyme Activity Levels of Human Milk

Based on a mixed-effect regression model adjusted for time effects based on the outcomes of repeated measurements, significantly higher SIgA levels (β-coefficient = 0.98, 95% CI = 0.57 to 1.40, *p* < 0.001) in human milk were observed after storage for six months in deep freezers (−22.3 °C) compared with upright freezers on top of refrigerators (−16.7 °C); however, there was no significant change in the corresponding lysozyme activity (β-coefficient = 0.02, 95% CI = −0.01 to 0.04, *p* = 0.136) (Table 4).

## 4. Discussion

We examined the effects of different storage conditions on the SIgA concentration and lysozyme activity of HM using a design that replicated domestic conditions. Compared with fresh HM, the SIgA concentration and lysozyme activity of HM stored in a refrigerator for four days decreased significantly. SIgA levels were stable during the first six months of freezing in either upright freezers on top of refrigerators or deep freezers, whereas the lysozyme activity significantly decreased after storage in either type of freezer. HM stored in a deep freezer had a higher SIgA concentration and lysozyme activity compared to in a refrigerator for four days or freezing in an upright freezer on top of a refrigerator for six months.

The temperatures of the refrigerators, upright freezers on top of refrigerators, and deep freezers in our study fell within a wide range of temperatures. This is consistent with prior research demonstrating that refrigerator temperatures vary widely. In 2002, Laguerre et al. monitored the temperatures inside domestic refrigerators in France and determined that the average temperature was 6.6 °C, with a minimum of 0.9 °C and a maximum of 11.4 °C [36]. In the previous decade, other surveys revealed relatively consistent results, with mean temperatures between 4.5 and 6.6 °C and maximum temperatures between 11 and 14 °C [37]. Recent research published in 2020 by Ovca et al. [38] showed that 20% of refrigerators had an average temperature below 4 °C, 30% had an average temperature between 4 and 6 °C, and 50% had an average temperature higher than 6 °C. Few studies have been conducted on domestic freezer temperatures. In a survey conducted in the United Kingdom in 2014 by Evan et al. [37], the average freezer temperature was −20.1 °C, the minimum mean was −41.1 °C, and the maximum mean was −11.1 °C. These findings suggest that domestic refrigerators and freezers operate across a wide range of temperatures, which may differ from the HM storage guidelines (Table 1). Home-based appliance settings and storage conditions should thus be considered in future research for the optimal infection control and quality of stored expressed HM.

We found that the level of SIgA in refrigerated HM decreased significantly after four days. This is in contrast with earlier studies which demonstrated that the SIgA concentrations were stable for two [25] or four days [27] in a refrigerator. Our results for SIgA concentration under deep freezing were similar to the majority of studies reporting that deep freezer storage (−20 °C) does not influence SIgA concentration over the course of one [24], six [25], or nine months [26]. However, an earlier study reported that a reduction in SIgA concentration in frozen HM stored at −20 °C for four weeks compared with fresh HM [29].

The inconsistency of these results may be due to differences in methodologies (Table 5). For example, some studies collected HM at the duration specified in the research protocol and then stored it at −80 °C until it was analyzed [25,27]. Freezing at extremely low temperatures (−80 °C) may affect the immune proteins in HM during the cooling and thawing processes [27] because of the degree of protein denaturation generated by the effects of temperature change via proteolysis, refolding, or recrystallization [39,40,41], resulting in inconsistent outcomes. Although the reports on the effect of storage conditions on the SIgA concentration were inconsistent, lysozyme levels have been repeatedly reported to significantly decrease after storage compared with fresh milk [24,26,30], similar to our study. These consistent findings may be explained by the fact that lysozymes are stable at an acidic pH but highly unstable at a neutral pH [42,43].

A strength of our study is that all HM samples were collected within three hours and were analyzed against baseline HM (fresh HM) within 24 h without freezing, which ensured the uniformity of the HM samples and minimized the effects of cooling and thawing on the baseline HM. However, this study has some significant limitations. First, the available resources determined the sample size. Second, we stored the samples in refrigerators and freezers in the laboratory, which limited the opening times and allowed the temperature to be maintained at a more stable level than under real household conditions. Third, we did not collect data on the proportion of mothers storing breast milk in refrigerators and freezers at home in Thailand. Fourth, we only investigated two immunological factors even though HM contains a variety of bioactive compounds. Therefore, future research is needed to determine the effects of home-based HM storage on a variety of bioactive compounds to identify the optimal storage conditions for expressed HM.

## 5. Conclusions

Our data support the superiority of fresh human milk over stored HM; accordingly, mothers should be encouraged to provide fresh HM to deliver the maximum protection to their infants. However, when fresh HM is not available, storage of human milk for six months in a deep freezer stabilizes the SIgA concentration and preserves the lysozyme activity more effectively than refrigeration for four days or freezing for six months in upright freezers. We suggest that support for lactating mothers to access optimal storage conditions should be incorporated into health policy initiatives aimed at promoting prolonged breastfeeding as well as maximizing the immunogenicity of expressed HM.

## Figures and Tables

**Figure 1 ijerph-19-13203-f001:**
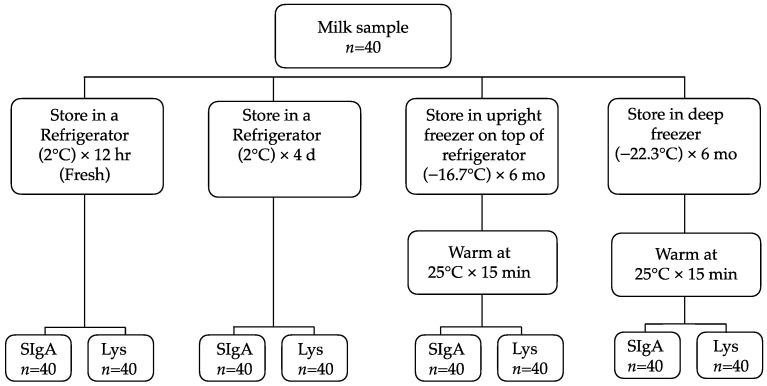
Diagram of the study protocol. Lys, lysozyme; SIgA, secretory immunoglobulin A.

**Figure 2 ijerph-19-13203-f002:**
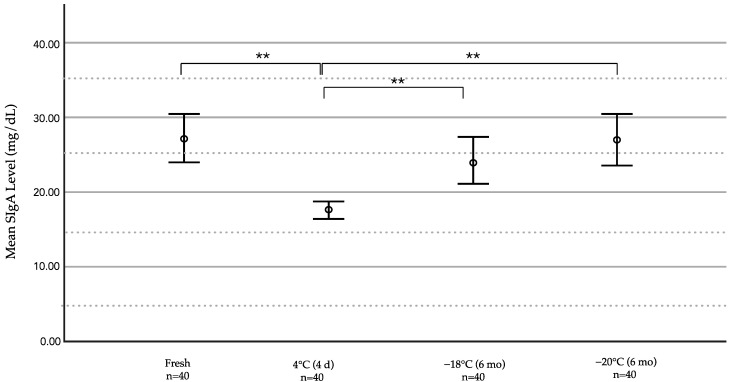
Comparison of SIgA concentrations between fresh, refrigerated (2 °C), frozen at −16.7 °C, and frozen at −22.3 °C HM samples. Data were analyzed using one-way analysis of variance (ANOVA) with Dunnett T3 pairwise comparisons for the SIgA concentration. ** *p* < 0.001.

**Figure 3 ijerph-19-13203-f003:**
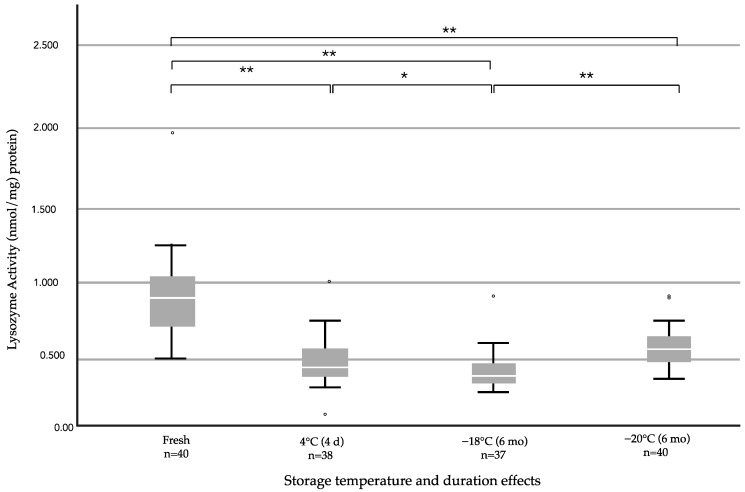
Comparison of lysozyme activities between fresh, refrigerated (2 °C), frozen at −16.7 °C, and frozen at −22.3 °C HM samples. The data were analyzed using Kruskal–Wallis test with Bonferroni correction for pairwise comparisons. The lysozyme activities are presented as the medians and percentiles (25th, 75th). * *p* < 0.05 and ** *p* < 0.001.

**Table 1 ijerph-19-13203-t001:** Summary of human milk storage guidelines.

Institute/Last Reviewed (Year)	Storage Duration and Temperature (°F/°C) for Freshly Expressed HM			
	Countertop	Refrigerator	Freezer	
			Freezer	Deep Freezer
Thai Ministry of Public Health/2022 [31]	3–4 h (80 °F/27 °C to 90 °F/32 °C)	3–5 d (32 °F/0 °C to 39 °F/4 °C)	3–6 m (0 °F/−18 °C or colder)	6–12 m (−4 °F/−20 °C or colder)
The American Academy of Pediatrics/2016 [32]	Up to 4 h (up to 77 °F/25 °C)	Up to 4 d (39 °F/4 °C)	Up to 9 m (0 °F/−18 °C or colder)	Up to 12 m (−4 °F/−20 °C or colder)
Australian Breastfeeding Association/2020 [33]	6–8 h (78.8 F/26 °C or lower)	Under 72 h; store at the back, where it is coldest	3 m in the freezer section of the refrigerator with a separate door (0 °F/−18 °C)	6–12 m (−4 °F/−20 °C)
The Academy of Breastfeeding Medicine/2017 [28]	4 h (60 °F/16 °C to 85 °F/29 °C)	4 d (39 °F/4 °C)	Under 6 m is best; up to 12 m is acceptable (25 °F/4 °C or colder)	
The Centers for Disease Control and Prevention/2022 [34]	Up to 4 h (77 °F/25 °C)	Up to 4 d (40 °F/4 °C)	Under 6 m is best; up to 12 m is acceptable (0 °F/−18 °C or colder)	

h, hours; d, days; m, months.

**Table 2 ijerph-19-13203-t002:** Maternal and infant characteristics (*n* = 40).

Characteristics	Mean ± SD or *n* (%)
Maternal	
Age (year)	28.6 ± 4.8
BMI (kg/m^2^)	23.8 ± 3.4
Birth order (first)	25 (62.5)
Delivery methods (vaginal birth)	29 (72.5)
Infant	
Age (month)	3.3 ± 0.3
Gestation age (week)	38.7 ± 1.0
Weight (kg)	3.1 ± 0.3

BMI, body mass index; the values are presented as the mean ± SD or percentage (%).

**Table 3 ijerph-19-13203-t003:** SIgA and lysozyme activity levels detected in fresh HM and HM samples frozen at −18 and −20 °C.

		Fresh Sample	Stored in a Refrigerator (2 °C) × 4 d	Frozen for 6 Months	
				Upright Freezer (−16.7 °C)	Deep Freezer (−22.3 °C)
SIgA (mg/dL)	*n*	40	40	40	40
	Mean ± SD (95% CI)	27.329 ± 10.185 (24.072, 30.586)	17.426 ± 4.674 (15.931, 19.921)	24.077 ± 9.296 (21.104, 27.050)	27.178 ± 9.690 (27.078, 30.0277)
	Minimum–Maximum	10.608–50.875	9.560–25.593	8.444–44.211	11.830–56.530
	Median	24.987	16.723	22.350	26.510
	Percentile (25th, 75th)	19.298, 34.900	13.693, 21.565	16.450, 28.628	19.120, 34.417
	% Decrease	-	36.23	11.90	0.60
Lysozyme (nmol/mg protein)	*n*	40	38	37	40
	Mean ± SD (95% CI)	0.923 ± 0.247 (0.844, 1.002)	0.459 ± 0.128 (0.417, 0.501)	0.391 ± 0.101 (0.358, 0.425)	0.564 ± 0.120 (0.525, 0.602)
	Minimum–Maximum	0.514–1.877	0.088–0.755	0.205–0.634	0.363–0.893
	Median	0.933	0.458	0.374	0.548
	Percentile (25th, 75th)	0.745, 1.077	0.376, 0.526	0.323, 0.440	0.489, 0.614
	% Decrease	-	44.64%	48.65%	38.89%

% Decrease = percentage decrease in the mean value compared to the fresh sample. SIgA, secretory immunoglobulin A.

**Table 4 ijerph-19-13203-t004:** Changes in SIgA concentrations and lysozyme activities of human milk, comparing storage in a deep freezer (−22.3 °C) and in an upright freezer on top of a refrigerator (−18.7 °C) over a six-month period.

Parameters	Study Groups	Adjusted β-Coefficient (Change in the Level per Month)	95% CI	*p*-Value
SIgA (mg/dL)	Deep freezer	0.98	0.57 to 1.40	<0.001
	Upright freezer	(ref.)		
Lysozyme activity (U/mg protein)	Deep freezer	0.02	−0.01 to 0.04	0.136
	Upright freezer	(ref.)		

Adjusted β-coefficient obtained from the mixed-effect regression model adjusted for time effects based on the outcomes of repeated measurements; upright freezer = upright freezer on top of the refrigerator; SIgA, secretory immunoglobulin A.

**Table 5 ijerph-19-13203-t005:** Overview of studies investigating the effects of storage conditions on SIgA concentration and lysozyme activity.

Ref.	Year	Sample (*n*)	Current LabCorp Method		Intervention	Effects		Comments
			SIgA	Lysozyme		SIgA Concentration	Lysozyme	
Kasiati et al. [30]	2021	11	NS	Enzyme-linked immunosorbent assay (ELISA; Fine Test Biotech, cat no. EH3314, China)	Comparison of HM stored at room temperature (26 °C), refrigerated (4 °C), and frozen (−20 °C).	NS	Lower at room temperature or when refrigerated than when frozen (*p* < 0.05).	HM stored at room temperature or refrigerated was analyzed at 1, 3, and 6 h (30–40 ng/mL), and frozen milk was analyzed at 1 (80–90 ng/mL), 3 (50–60 ng/mL), and 6 (30–40 ng/mL) days.
Ahrabi et al. [26]	2016	40	Enzyme-linked immunosorbent assay (ALPACO Diagnostics, Salem, NH, USA)	NS	Comparison of fresh, frozen (immediate storage at −20 °C), and refrigerated frozen HM (storage for 72 h at 4 °C before −20 °C) with the baseline HM.	SIgA of fresh and refrigerated frozen HM was stable for up to 9 months.	NS	A baseline (time 0) aliquot was held at −80 °C until analysis. Samples were removed from the freezer (−20 °C) at 1, 3, 6, and 9 months and held at −80 °C until analyzed.
Chang et al. [24]	2013	Fresh milk (14); frozen milk (Milk Bank)	SIgA ELISA kit (K8870; Immundiagnostik AG, Bensheim, Germany)	Lysozyme enzyme immunoassay kit (Biomedical Technologies Inc., Stoughton, MA, USA)	Comparison of frozen HM (−20 °C) with fresh HM.	Stable	Reduced (*p* < 0.0001)	Fresh HM was stored at 4 °C and processed within 24 h (IgA: 1000–1200 μg/mL, lysozyme: 60–70 μg/mL). Frozen HM was stored at −20 °C for up to 4 weeks before analysis (IgA: 1000–1200 μg/mL, lysozyme: 30–40 μg/mL).
Santana et al. [25]	2012	10 (cooling storage), 2 or 3 donors (pooling milk from frozen milk)	ELISA quantitation kit from Bethyl Laboratories (Montgomery, TX, USA)	NS	Comparison of refrigerated HM (4 °C) and frozen HM (−20 or −80 °C) with baseline HM.	SIgA of refrigerated and frozen HM was stable for 48 h and 6 months, respectively; SIgA of frozen HM (12 months) was reduced (*p* < 0.01).	NS	A baseline (time 0) aliquot was stored at −80 °C until analysis (8.8 mg/mL). Refrigerated HM was removed from the refrigerator at 6 (8.6 mg/mL), 12 (8.2 mg/mL), 24 (8.4 mg/mL), and 48 (8.6 mg/mL) h. Frozen HM was removed from the freezer at 6 and 12 months before held at −80 °C until analysis.
Slutzah et al. [27]	2010	36	Enzyme-linked immunosorbent assay (ALPACO Diagnostics, Salem, NH, USA)	NS	Comparison of refrigerated HM (4 °C) with baseline HM.	IgA of refrigerated HM was stable for up to 96 h.	NS	A baseline (time 0) aliquot was stored at −80 °C until analysis. HM was removed from the refrigerator at 24, 48, 72, and 96 h and held at −80 °C until analysis.
Akinbi et al. [29]	2010	18 (fresh); 20 (frozen)	Enzyme-linked immunosorbent assay (ALPACO Diagnostics, Salem, NH, USA)	Antihuman lysozyme (Accurate Chemical and Scientific Corp., New York, NY, USA)	Comparison of frozen HM (−20 °C) with fresh HM.	Reduced (*p* < 0.0001)	Reduced (*p* < 0.001)	Frozen HM was stored at −20 °C for 4 weeks.

NS, not studied.

## Data Availability

The data presented in this study are available on request from the corresponding author.

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
