# Peer review of "Comparison of Effects of Storage at Different Temperatures in a Refrigerator, Upright Freezer on Top of Refrigerator, and Deep Freezer on the Immunoglobulin A Concentration and Lysozyme Activity of Human Milk"

_ijerph, 2022, doi:10.3390/ijerph192013203_

Round 1

Reviewer 1 Report

This study examine different storage conditions (at 2-4 C, -18 C and -80 C) to determine the secretory immunoglobulin A (SIgA) concentration and lysozyme activity. Study is important for human milk but instead of using assay kits  HPLC might be used for quantitatively more certain results and more novel study. My recommendations are below:

-Similarity was found 34% (by Turnitin) which is so high for scientific papers. It should be revised.

-Table 1 can be written as text under introduction

-The size of table 2 is different

Author Response

Response to Reviewer 1:

We are pleased to submit the revised manuscript entitled: "The Effect of Storing Expressed Human Milk under Refrigerator, Upright Freezer on Top of Refrigerator, and Deep Freezer Temperatures on the Immunoglobulin A Concentration and Lysozyme Activity" that we have changed from the original title "The Effect of Storing Expressed Human Milk under Domestic Conditions on the Immunoglobulin A Concentration and Lysozyme Activity." We would like to thank the reviewers and editor for their precious time and invaluable comments. We have carefully addressed all the comments, especially the comment that mentions the duplication rate is quite high (reviewer 1, comment 1). We have rewritten revision manuscripts and undergo English language editing by MDPI (Specialist) to improve our English and reduce duplication from 34% to 25%. The corresponding changes and refinements made in the revised manuscript are summarized in our responses below.

Overall: This study examine different storage conditions (at 2-4 °C, -18 °C and -80 °C) to determine the secretory immunoglobulin A (SIgA) concentration and lysozyme activity. Study is important for human milk but instead of using assay kits HPLC might be used for quantitatively more certain results and more novel study. My recommendations are below:

Response: We thank the reviewer for the valuable comment. These comments are very constructive and will help us to improve the manuscript and our future work. We have rewritten the conclusion as the reviewer evaluate in the conclusion can be improved. Please see our revisions as follows:

in page 1, lines 31-34;

“If HM is to be stored, then storage in a deep freezer is potentially a more effective method for preservation of SIgA concentrations and lysozyme activity than storage by refrigeration for four days or in an upright freezer on top of a refrigerator for six months.”

Q 1: Similarity was found 34% (by Turnitin) which is so high for scientific papers. It should be revised.

A 1: We have rewritten revision manuscripts and undergone English language editing by MDPI (Specialist) to improve our English and reduce duplication from 34% to 25% (by Turnitin). The majority of the similar are technical terms in the Materials and Methods that may occur since we recruited the participant and used the baseline sample in the same procedure as our previously published article that focused on the impact of SIgA concentration and lysozyme activity on “thawing rate” and “warming temperature” on the sample that was frozen for two months (this manuscript used the refrigerated sample at four days, and frozen at six months), which we have cited in reference number 35.

Q 2: Table 1 can be written as text under introduction

A 2: We have written to explain Table 1 as text. We also have moved table 1 from the introduction to the discussion (now Table 5 in the revised manuscript) and added more information as suggested by another reviewer. Please see our revisions as follows:

in page 2, lines 73-79;

“For example, some studies collected and compared SIgA baseline data from individual HM samples with those from frozen milk samples obtained by pooling HM from a milk bank [24,25]. Whereas, Ahrabi et al. [26] compared the IgA level in frozen samples with baseline samples from the same group of participants. In addition, in previous studies, milk sample have been collected at different times and stored in a laboratory freezer (-80 °C) until analysis, causing each sample to be exposed to extremely low temperatures for varying amounts of time before they were analyzed [25-27].”

Q 3: The size of table 2 is different

A 3: We have changed the size in Table 2 (now Table 1 in the revised manuscript) and the remaining sections.

Reviewer 2 Report

Thanks researchers for conducting this interesting research. I read the article with great interest. Few comments from me:

Introduction part could be improved - maybe authors can provide additional information - how much women (in %) in Thailand are breastfeeding, how much babies are receiving human milk for the first 6, 12 months of life?

Section 2.3. Milk Collection and Acquisition of Milk Samples:

It is not quite clear, overall how many mL of human milk were collected in total from one participant. Authors write that milk was poured into a 50-mL polypropylene centrifuge tube and in the end eight 10 mL aliquots were provided.

Have authors taken into consideration if other parameters like maternal age, time postpartum, delivery mode etc. can affect the concentration of SIgA and lysozyme acitivity?

Have authors collected data if participating women were making milk banks at home (storing/freezing milk)? Is it overall popular in Thailand for breastfeeding mothers to make milk banks at home, storing milk in freezers?

For Discussion part:

Maybe authors can compare in the Table obtained data for concentration of SIgA and lysozyme acitivity in different storage with data from other counties? It would make clearer to compare the data.

Author Response

Response to Reviewer 2:

We are pleased to submit the revised manuscript entitled: "The Effect of Storing Expressed Human Milk under Refrigerator, Upright Freezer on Top of Refrigerator, and Deep Freezer Temperatures on the Immunoglobulin A Concentration and Lysozyme Activity" that we have changed from the original title "The Effect of Storing Expressed Human Milk under Domestic Conditions on the Immunoglobulin A Concentration and Lysozyme Activity." We would like to thank the reviewers and editor for their precious time and invaluable comments. The corresponding changes and refinements made in the revised manuscript are summarized in our responses below.

Overall: Thanks researchers for conducting this interesting research. I read the article with great interest. Few comments from me:

Response: We thank the reviewer for the valuable comment. These comments are very constructive and will help us to improve the manuscript.

Q 1: Introduction part could be improved - maybe authors can provide additional information - how much women (in %) in Thailand are breastfeeding, how much babies are receiving human milk for the first 6, 12 months of life?

A 1: We have added more information about the success rate of breastfeeding in Thailand. Please see our revisions as follows:

in page 2, lines 52-57;

“According to the 2019 Thailand Multiple Indicator Cluster Survey, the proportion of exclusive and predominant breastfeeding for six months was 14% and 31%, respectively. The proportion of children aged between 13 and 15 months who are breastfed is 25%, and for those between 20 and 23 months it is 15%. This report includes all methods of breastmilk feeding, namely direct nursing, expressing, and bottle or cup feeding [14].”

Q 2: Section 2.3. Milk Collection and Acquisition of Milk Samples:

It is not quite clear, overall how many mL of human milk were collected in total from one participant. Authors write that milk was poured into a 50-mL polypropylene centrifuge tube and in the end eight 10 mL aliquots were provided.

A 2: We have added more information, explaining that we collect a sample of one mother using three 50-mL polypropylene centrifuge tubes. Please see our revisions as follows:

in page 3, lines 127-129;

“The samples (120 mL) were then poured into three 50 mL polypropylene centrifuge tubes (NuncTM®, Roskilde, Denmark), of which 80 mL was analyzed for this study.”

Q 3: Have authors taken into consideration if other parameters like maternal age, time postpartum, delivery mode etc. can affect the concentration of SIgA and lysozyme acitivity?

A 3: Yes, while developing a research protocol, we consider the variables that may influence the SIgA concentration and lysozyme activity. Following is a brief summary of our review and a description of how we regulate these variables in our study.

1. The factors have been reported to affect human milk SIgA levels and lysozyme activity significantly.

1.1 The lactation stage; Colostrum contained more SIgA than transitional and mature milk, and the differences between traditional and mature milk IgA were inconclusive. The IgA concentration in mature milk was relatively stable in the first year and increased significantly in the following years [1,2]. In contrast to IgA levels, lysozyme activity was negatively associated with colostrum compared to transitional or mature milk. Lysozyme activity increases progressively, beginning about six months after delivery [3]. Therefore, we recruited lactating mothers who had given birth to infants aged one to six months for this study. 

1.2 Gestational age (Preterm delivery); SIgA concentration in human milk of preterm of both very preterm (<28–30 weeks) and preterm (30–36 weeks) milk is significantly higher than in term milk [4]. In terms of lysozyme activity, very preterm milk (28–32 weeks) had significantly higher lysozyme activity than term milk [5]. Therefore, we recruited lactating mothers who had given full infants for this study. 

1.3 Maternal underlying disease; SIgA concentration in human milk was reduced in some maternal conditions, such as gestational diabetes [6] and inflammatory bowel disease[7]. Therefore, we recruited healthy lactating mothers.

1.4 HM's SIgA levels and lysozyme activity can be influenced by the thawing process and heat treatment [8-10]. Therefore, we limit freeze-thaw effects by collecting milk samples from all participants on the same day in adequate amounts that can be aliquoted and stored under all storage conditions per the protocol. Consequently, each sample undergoes one cycle of freezing and thawing.

2. The factors have been reported to affect human milk SIgA levels and lysozyme activity non significantly.

2.1 SIgA and Lysozyme activity levels were not significantly affected by the mode of delivery [5]. 

2.2 Maternal age; There is no association between levels of SIgA immunoglobulin and maternal age [11,12]. However, we recruited mothers aged between 18-40 years for ethical reasons.

3. The factors that limit information regarding the impact on SIgA levels and lysozyme activity of HM but have been documented the impact on other components of HM.

3.1 Diurnal variation

3.2 The differences between fore milk and hind milk

Little is known about the effect of diurnal variation and differences between fore and hind milk on HM SIgA levels and lysozyme activity. However, they have documented the effect on other human milk components[13,14]. However, we have managed our studies by putting up more effort to ensure milk samples are as uniform and feasible as possible. Therefore, the pump was left on for approximately 15 minutes, or until no additional milk was expressed for at least 5 minutes, and all breast milk samples were collected between 8:00 a.m. and 11:00 a.m. to ensure that we have samples of both fore milk and hind milk and to minimize diurnal variation.

References:

  1. Czosnykowska-Łukacka, M.; Lis-Kuberka, J.; Królak-Olejnik, B.; Orczyk-Pawiłowicz, M. Changes in Human Milk Immunoglobulin Profile During Prolonged Lactation. Front Pediatr 2020, 8, 428, doi:10.3389/fped.2020.00428.
  2. Rio-Aige, K.; Azagra-Boronat, I.; Castell, M.; Selma-Royo, M.; Collado, M.C.; Rodríguez-Lagunas, M.J.; Pérez-Cano, F.J. The Breast Milk Immunoglobulinome. Nutrients 2021, 13, doi:10.3390/nu13061810.
  3. Riordan, J. The Biological Specificity of Breastmilk. In Breastfeeding and Human Lactation; Jones & Bartlett Learning: Burlington, MA, 2016; pp. 121-169.
  4. Underwood, M.A. Human milk for the premature infant. Pediatr Clin North Am 2013, 60, 189-207, doi:10.1016/j.pcl.2012.09.008.
  5. Trend, S.; Strunk, T.; Lloyd, M.L.; Kok, C.H.; Metcalfe, J.; Geddes, D.T.; Lai, C.T.; Richmond, P.; Doherty, D.A.; Simmer, K.; et al. Levels of innate immune factors in preterm and term mothers' breast milk during the 1st month postpartum. Br J Nutr 2016, 115, 1178-1193, doi:10.1017/s0007114516000234.
  6. Smilowitz, J.T.; Totten, S.M.; Huang, J.; Grapov, D.; Durham, H.A.; Lammi-Keefe, C.J.; Lebrilla, C.; German, J.B. Human milk secretory immunoglobulin a and lactoferrin N-glycans are altered in women with gestational diabetes mellitus. J Nutr 2013, 143, 1906-1912, doi:10.3945/jn.113.180695.
  7. Meng, X.; Dunsmore, G.; Koleva, P.; Elloumi, Y.; Wu, R.Y.; Sutton, R.T.; Ambrosio, L.; Hotte, N.; Nguyen, V.; Madsen, K.L.; et al. The Profile of Human Milk Metabolome, Cytokines, and Antibodies in Inflammatory Bowel Diseases Versus Healthy Mothers, and Potential Impact on the Newborn. J Crohns Colitis 2019, 13, 431-441, doi:10.1093/ecco-jcc/jjy186.
  8. Akinbi, H.; Meinzen-Derr, J.; Auer, C.; Ma, Y.; Pullum, D.; Kusano, R.; Reszka, K.J.; Zimmerly, K. Alterations in the host defense properties of human milk following prolonged storage or pasteurization. J Pediatr Gastroenterol Nutr 2010, 51, 347-352, doi:10.1097/MPG.0b013e3181e07f0a.
  9. Escuder-Vieco, D.; Espinosa-Martos, I.; Rodríguez, J.M.; Fernández, L.; Pallás-Alonso, C.R. Effect of HTST and Holder Pasteurization on the Concentration of Immunoglobulins, Growth Factors, and Hormones in Donor Human Milk. Front Immunol 2018, 9, 2222, doi:10.3389/fimmu.2018.02222.
  10. Evans, J.F., A. M.; Brown, T. Temperature control in domestic refrigerators and freezers. In Proceedings of the 3rd IIR International Cold Chain Conference, Twickenham, United Kingdom, 2014; p. 8.
  11. Bachour, P.; Yafawi, R.; Jaber, F.; Choueiri, E.; Abdel-Razzak, Z. Effects of smoking, mother's age, body mass index, and parity number on lipid, protein, and secretory immunoglobulin A concentrations of human milk. Breastfeed Med 2012, 7, 179-188, doi:10.1089/bfm.2011.0038.
  12. Islam, S.K.; Ahmed, L.; Khan, M.N.; Huque, S.; Begum, A.; Yunus, A.B. Immune components (IgA, IgM, IgG, immune cells) of colostrum of Bangladeshi mothers. Pediatr Int 2006, 48, 543-548, doi:10.1111/j.1442-200X.2006.02291.x.
  13. Chung, M.Y. Factors affecting human milk composition. Pediatr Neonatol 2014, 55, 421-422, doi:10.1016/j.pedneo.2014.06.003.
  14. Geraghty, S.R.; Davidson, B.S.; Warner, B.B.; Sapsford, A.L.; Ballard, J.L.; List, B.A.; Akers, R.; Morrow, A.L. The development of a research human milk bank. J Hum Lact 2005, 21, 59-66, doi:10.1177/0890334404273162.

Q 4: Have authors collected data if participating women were making milk banks at home (storing/freezing milk)?

A 4: We did not collect information on whether or not participating women made breast milk banks at home. We have mentioned this point in the limitation and provided more information from the previous article regarding this issue. Please see our revisions as follows:

in page 13 line 345-346 as follows

“Third, we did not collect data on the proportion of mothers storing breastmilk in refrigerators and freezers at home in Thailand.”

Q 5: Is it overall popular in Thailand for breastfeeding mothers to make milk banks at home, storing milk in freezers?

A 5: We could not find the document that indicated breastfeeding mothers in Thailand making milk banks at home and storing the milk in freezers. However, from an earlier article, we added more related information about mothers making milk banks at home. Please see our revisions as follows:

in page 2, lines 57-59;

 "According to studies conducted in Perth, Australia, the number of mothers who expressed breast milk increased from 38% to 69% between 1992 and 2002 [9]."

Q 6: For Discussion part: Maybe authors can compare in the Table obtained data for concentration of SIgA and lysozyme acitivity in different storage with data from other counties? It would make clearer to compare the data.

A 6: We include data for a concentration of SIgA and lysozyme activity in different storage with data from other counties in table 5 in the discussion. The concentration of SIgA and lysozyme activity was presented in only three out of six studies (Kasiati et al., Chang et al., Santana et al.) due to the limited number of studies that report exact values of the concentration of SIgA and lysozyme activity. In table 5, one of the studies reported changing proportion after storage (Akinbi et al.), and two reported text to descript nonsignificant change after storage (Ahrabi et al., Slutzah et al.). Please see our revisions in table 5.

Reviewer 3 Report

The title of the manuscript is inconsistent with the research carried out because the authors stored the samples on their own and not in the nursing mother's home conditions. The storage temperature is the correct factor, not the Domestic Conditions for the storage of human milk. The work concerns the influence of cooling and freezing temperature on the content of immonoglobulin A and lysozyme in human milk. Unfortunately, I was disappointed with the Authors' honesty, because they have already published these research results in another journal:

Xuejing LiPenprapa SivirojJetsada RuangsuriyaNitthinan Yousaibua & Krongporn Ongprasert . Effects of the thawing rate and heating temperature on immunoglobulin A and lysozyme activity in human milk. International Breastfeeding Journal ,volume 17, Article number: 52 (2022).

Therefore, I believe that the work should not be published in IJERPH.

Author Response

Response to Reviewer 3:

We are pleased to submit the revised manuscript entitled: "The Effect of Storing Expressed Human Milk under Refrigerator, Upright Freezer on Top of Refrigerator, and Deep Freezer Temperatures on the Immunoglobulin A Concentration and Lysozyme Activity" that we have changed from the original title "The Effect of Storing Expressed Human Milk under Domestic Conditions on the Immunoglobulin A Concentration and Lysozyme Activity." We would like to thank the reviewers and editor for their precious time and invaluable comments. We have carefully addressed all the comments, especially the comment that mentions a similar result has been published in a previous article (reviewer 3, comment 2). The corresponding changes and refinements made in the revised manuscript are summarized in our responses below.

Q 1: The title of the manuscript is inconsistent with the research carried out because the authors stored the samples on their own and not in the nursing mother's home conditions. The storage temperature is the correct factor, not the Domestic Conditions for the storage of human milk.  

A 1: We have changed the original title (The Effect of Storing Expressed Human Milk under Domestic Conditions on the Immunoglobulin A Concentration and Lysozyme Activity) to the new title in the revised manuscript by not mention on “Domestic Conditions” as the suggested by reviewer 3.

Please see our revisions as follows:

in page 1, lines 2-4;

The Effect of Storing Expressed Human Milk under Refrigerator, Upright Freezer on Top of Refrigerator, and Deep Freezer Temperatures on the Immunoglobulin A Concentration and Lysozyme Activity

Q 2: The work concerns the influence of cooling and freezing temperature on the content of immonoglobulin A and lysozyme in human milk. Unfortunately, I was disappointed with the Authors' honesty, because they have already published these research results in another journal:

Xuejing Li, Penprapa Siviroj, Jetsada Ruangsuriya, Nitthinan Yousaibua & Krongporn Ongprasert . Effects of the thawing rate and heating temperature on immunoglobulin A and lysozyme activity in human milk. International Breastfeeding Journal ,volume 17, Article number: 52 (2022).

A 2: I absolutely realize how my mistake disappointed the reader who spent valuable time reviewing my work. I deeply apologize for my mistake. I have learned an important lesson from my mistakes, and I appreciate the reader for pointing this out. We have defined the data that repeated from the previously published article (same participants and baseline concentration of IgA and Lysozyme) and cited our earlier article in the method.

Please see our revisions as follows:

in page 4, lines 108-110;

 “Details on the sample size, participant recruitment, milk collection, and acquisition of milk samples have been published elsewhere [35].”

in page 4, lines 129-130;

 “and the remaining 40 mL had been used for analysis in our earlier study[35].”

in page 5, lines 135-137;

“Two 10 mL aliquots were refrigerated and analyzed within 24 hours to identify the baseline SIgA levels and lysozyme activity (fresh), and these baseline results have been published elsewhere [35].”

Except for the same participant was recruited, and the same baseline sample result was used, other content were major difference. I would like to highlight six significant differences between the previous article and this manuscript.

1. The research questions:

- The previous article "Is the changing temperature during the thawing process and warming temperature of frozen HM affect IgA level and Lysozyme activity?"

- This manuscript "How does storage in domestic temperature and duration affect IgA concentration and Lysozyme level in HM?"

2. The objectives:

- The previous article "to identify optimal thawing methods and the feeding temperatures (warming temperature) of frozen HM (-18 °C) to preserve IgA concentration and Lysozyme level in HM."

- This manuscript "to examine the effect of storage temperature that the general public may access for use in their homes, including refrigerators, upright freezers on top of refrigerators, and deep freezers (chest freezers).”

3. The methods:

- The previous article "We compared two thawing methods, slow thawing and rapid thawing (placing the container overnight in a refrigerator at four °C before warming v.s. immediately thawing in warm water after removing the sample from the freezer) and two feeding temperatures (room temperature and physiological temperature (25 °C v.s. 37 °C).”

- This manuscript " We compared three storage temperatures (refrigerators, upright freezers on top of refrigerators, and deep freezers).”

4. The storage duration:

- The previous article “Frozen for two months

- This manuscript "Refrigerated four days or frozen for six months"

  1. Statistical analyses:

- The previous article “The SIgA concentrations were compared by one-way analysis of variance (ANOVA) with Tukey’s HSD (honest significance) pair-wise comparisons for parametric testing, and lysozyme activities were compared using the Kruskal–Wallis test with Dwass-Steel-Critchlow-Fligner pairwise comparisons for nonparametric testing.”

- This manuscript “One-way analysis of variance with Dunnett T3 and the Kruskal–Wallis test with Bonferroni correction were used for pairwise comparisons of the SIgA concentration and lysozyme activity.”

6. The main result:

- The previous article "Slow thawing (placing the container overnight in a refrigerator at 4 °C before warming) preserved higher SIgA concentrations and lysozyme activity than rapid thawing at 37 °C, but the difference was not significant."

- This manuscript "SIgA concentration and lysozyme activity of HM stored in the refrigerator for four days and freezers for six months were significantly lower than those of fresh HM (p <0.001). During the first six months of storage in both types of freezers, the SIgA levels were stable, whereas the lysozyme activity significantly decreased (p <0.001)."

Round 2

Reviewer 1 Report

Desired corrections were made to improve the quality of the publication.

Author Response

Response to Reviewer 1:

We would like to thank the editors and reviewers for their time and thoughtful comments on our manuscript. Because the rating in the "Are the conclusions supported by the results?" are not displayed, we tried to improve our conclusions by removing the part that did not directly relate to our study. We have rewritten the conclusion. Please see our revisions as follows:

in page 12, lines 347-354;

“We suggest that support for lactating mothers to access optimal storage conditions should be incorporated into health policy initiatives aimed at promoting prolonged breastfeeding as well as maximizing the immunogenicity of expressed HM.

Reviewer 3 Report

The authors put a lot of work into improving this manuscript and reduced some duplicate studies from another published manuscript. However, 25% or 1/4 of the manuscript is too many duplicate borrowings. The authors should further reduce this score.

Author Response

Response to Reviewer 3:

We are pleased to submit the revised manuscript entitled: "The Effect of Storing Expressed Human Milk under Refrigerator, Upright Freezer on Top of Refrigerator, and Deep Freezer Temperatures on the Immunoglobulin A Concentration and Lysozyme Activity. We would like to thank the editors 

and reviewers for their time and consideration in the careful review of this revised manuscript. Our response is reported below.

Q 1: The authors put a lot of work into improving this manuscript and reduced some duplicate studies from another published manuscript. However, 25% or 1/4 of the manuscript is too many duplicate borrowings. The authors should further reduce this score.

A 1: We have rewritten revision manuscripts and reduced duplication from 25% to 21%. We have attached the pdf file of the report 21% by Turnitin after we rewritten and removed the section which did not include the main text (author information, ethical considerations, funding, and acknowledgments), which is almost the same condition as our previous study on thawing and warming rate already cited in this manuscript (reference number 35). We would like to point out that the most remaining duplicate was not a sentence or phrase but standard chemistry laboratory equipment naming, technical terms in statistic analysis, and "SIgA concentrations and lysozyme activity." Please see our report by Turnitin in the attached file (name Turnitin_Round2_Frozen).
